# Uniformly Dispersed Sb-Nanodot Constructed by In Situ Confined Polymerization of Ionic Liquids for High-Performance Potassium-Ion Batteries

**DOI:** 10.3390/molecules28135212

**Published:** 2023-07-05

**Authors:** Cunliang Zhang, Zhengyuan Chen, Haojie Zhang, Yanmei Liu, Wei Wei, Yanli Zhou, Maotian Xu

**Affiliations:** 1Henan Key Laboratory of Bimolecular Reorganization and Sensing, Henan Engineering Center of New Energy Battery Materials, School of Chemistry and Chemical Engineering, Shangqiu Normal University, Shangqiu 476000, China; zcliang@126.com (C.Z.); weiweizzuli@163.com (W.W.); 2School of Petrochemical Engineering, Liaoning Petrochemical University, Fushun 113001, China; 3Department of Public Science, Shangqiu Medical College, Shangqiu 476000, China

**Keywords:** Sb, ionic liquids, nanodot, anode, potassium-ion batteries

## Abstract

Antimony (Sb) is a potential candidate anode for potassium-ion batteries (PIBs) owing to its high theoretical capacity. However; in the process of potassium alloying reaction; the huge volume expansion (about 407%) leads to pulverization of active substance as well as loss of electrical contact resulting in rapidly declining capacity. Herein; uniformly dispersed Sb-Nanodot in carbon frameworks (Sb-ND@C) were constructed by in situ confined polymerization of ionic liquids. Attributed to the uniformly dispersed Sb-ND and confinement effect of carbon frameworks; as anode for PIBs; Sb-ND@C delivered a superior rate capability (320.1 mA h g^−1^ at 5 A g^−1^) and an outstanding cycling stability (486 mA h g^−1^ after 1000 cycles; achieving 89.8% capacity retention). This work offers a facile route to prepare highly dispersed metal-Nanodot via the in situ polymerization of ionic liquid for high-performance metal-ion batteries

## 1. Introduction

With the gradual application of lithium-ion batteries in large-scale electrochemical energy storage devices, the demand for lithium will be gradually on the rise. However, the low content (only 0.0017%) and uneven distribution of lithium in the earth’s crust will have a significant impact on the price of lithium-ion batteries [1]. Therefore, there is an urgent need for the development of an alternative system that is low cost and abundant in reserves. In recent years, potassium-ion batteries (PIBs) have gradually come into the spotlight in the field of next-generation energy storage, as to the high natural K abundance of 2.09 wt.% and close K/K^+^ redox potential of −2.93 V (vs. −3.04 V for Li/Li^+^) [2,3]. K^+^ is relatively difficult to extract and insert from the electrode materials due to the larger radius of K^+^ (1.38 Å) than that of Li^+^ (0.76 Å). Large volume changes lead to electrode pulverization which shortens life, decreases capacity and deteriorates rate performance during the charge-discharge processes. As a consequence, it is of great significance for PIBs to explore and design appropriate electrode materials with inhibition of large-volume expansion and acceleration of K^+^ transport.

Great achievements have been made in the research of cathode materials for PIBs, whose performance is close to that of LIBs [4,5,6,7,8,9]. Compared with the current anode materials used in potassium ion batteries, carbon-based materials are difficult to meet the needs of practical applications due to their low specific capacity (less than 280 mA h g^−1^) [10,11,12]. Recently, the interest in alloy materials has increased because of their suitable potential and high theoretical specific capacity. Such as antimony (Sb) exhibits a high theoretical specific capacity of 660 mA h g^−1^ as the most promising electrode material for PIBs. However, the huge volume expansion (about 407%) in the process of potassium alloying reaction to form K_3_Sb leads to pulverization of active substances as well as loss of electrical contact which results in capacity decay rapidly even with poor cycling performance.

In order to overcome this obstacle, many schemes have been designed to alleviate its volume expansion and maintain the integrity of structure during the repeated insertion and extraction of bulky K^+^. In summary, they can be divided into two categories. On the one hand, design the structure and morphology at the nanoscale, while this kind of method is not suitable for large-scale production due to its harsh preparation conditions [13,14,15,16]. On the other hand, the carbon material as a buffer layer to maintain the structural stability of the material and alleviated volume expansion [14,15,16,17,18,19]. However, the introduction of carbon material reduces the energy density of the composites.

Hence, the development of a method for the synthesis of Sb particles with high loading and dispersion is a necessity for Sb-based materials. Ionic liquids (ILs) are organic molten salts composed entirely of anions and cations [20]. Ionic liquids can be used not only as green and nonvolatile solvents, but also for material dispersion due to their special charged properties and structural diversity [21]. Relying on the structural characteristics of ILs, we propose a concept to prepare highly dispersed Sb-Nanodot via the in situ polymerization of ionic liquid 1-vinyl-3-butyl imidazolium bromide ([VBIm]Br). SbCl_3_ was firstly dissolved in [VBIm]Br to form a homogeneous solution, then crosslinker divinylbenzene (DVB) and free radical initiator azobisisobutyronitrile (AIBN) were added for in situ polymerization, and Sb^3+^ ions were uniformly anchored in the in situ formed poly(ionic liquid) frameworkss. After pyrolysis under a reduced atmosphere, uniformly dispersed Sb-nanodot supported by nitrogen-doped carbon frameworks were obtained, as shown in Figure 1.

## 2. Experimental

### 2.1. Materials

Antimony trichloride (SbCl_3_, 99%), AIBN and ethyl alcohol were purchased from Aladdin Reagents. DVB was purchased from J&K Scientific (Beijing, China). [VBIm]Br was supplied by Lanzhou GreenChem ILs. Ar/H_2_ was acquired from Longhai Factory with a purity of 99.999%. DVB was purified to remove the inhibitor by distillation under reduced pressure. Other chemicals and solvents were used directly without further purification.

### 2.2. Preparation of Sb-Nanodot@C (Sb-ND@C)

Sb-Nanodot was prepared via ionic liquid polymerization. In a typical synthesis, an amount of 1.30 g of DVB, 1.16 g of [VBIm]Br and 2.30 g of SbCl_3_ were dissolved in 10 mL of ethyl alcohol under magnetic stirring. Afterward, 250 mg of AIBN was added into the transparent solution under slow stirring at 80 °C for 12 h. Antimony ions anchored in polymer gel were annealed under an Ar/H_2_ (vol, 95:5) atmosphere at 650 °C for 3 h with a heating rate of 5 °C min^−1^ to obtain Sb-ND@C.

### 2.3. Preparation of C

C was synthesized without SbCl_3_ by the same method as Sb-ND@C.

### 2.4. Preparation of Sb/rGO

The GO was dispersed in de-ionized water with a concentration of 1 mg mL^−1^, to which SbCl_3_ was added. The solution was kept stirring at 70 °C in a water bath for two hours. Afterward, the product was collected by vacuum filtration and freeze-dried, which was annealed under an Ar/H_2_ (vol, 95:5 atmosphere at 650 °C for 3 h with a heating rate of 5 °C min^−1^ to obtain Sb/rGO.

### 2.5. Characterizations

The crystal structure of the sample was surveyed by powder X-ray diffraction (XRD) on Bruker D8 advance equipped Cu Kα radiation (λ = 1.5406 Å) at a scan rate of 5° min^−1^ in the 2θ range 10–80°. Raman spectra of the sample were acquired by the Thermo Fisher DXR2xi Raman spectrometer at a 532 nm excitation wavelength. Infrared spectroscopy was performed on KBr pellets by a Fourier transform infrared (FT-IR) spectrometer (Thermo-Nicolet Nexus Spectrum 670). The information of chemical compositions and elemental states was collected by X-ray photoelectron spectrometer (XPS, Thermo Escalab 250Xi) with a monochromatic Al Kα X-ray source (1486.6 eV) operating at 150 W. The morphology of the samples was characterized by scanning electron microscopy (SEM) operating on a Thermo Scientific Verios G4 UC microscope equipped with an Ultim Max 170 (Oxford, UK) detector to detect the elemental distribution. Transmission electron microscopic (TEM) and high-resolution TEM (HRTEM) images were acquired using FEI Talos F200S instrument at an acceleration voltage of 200 kV.

### 2.6. Electrochemical Performances

The sample of Sb-ND@C were mixed with a binder carboxymethyl cellulose (CMC) and acetylene black with a weight ratio of 8:1:1 to prepare a homogeneous slurry. Adjusting the slurry concentration by adding deionized water. The slurry was coated on copper foil uniformly then dried under vacuum at 60 °C for 12 h. The mass loadings of the Sb-Nanodot in the working electrode were in the range of 1.00 ± 0.02 mg/cm^2^. The 2032 coin-type cell was assembled by a polypropylene glass microfiber filter (Whatman GF/D) as separator, potassium foil as counter electrode, Sb-ND@C as cathode in the glove box with an Ar atmosphere (H_2_O < 0.1 ppm, O_2_ < 0.1 ppm). The electrolyte was made of KFSI (3M KFSI in DME) solution. The galvanostatic charge-discharge measurements of Sb-ND@C were tested on Land test system (CT 2001A). Cyclic voltammetry (CV) was conducted on electrochemical workstation (Zennium, IM6, Kronach, Germany). CV measurements were performed in scan rates from 0.2 to 1.0 mV s^−1^ at potential ranges of 0.01 to 2.0 V.

## 3. Results and Discussion

The design concept of Sb-ND@C was illustrated in Figure 1a. DVB, [VBIm] Br and SbCl_3_ were dissolved into homogenous solution and polymerized into a gel by AIBN as initiator, in which the Sb precursors were confined at molecular level, as shown in Figure 1a. Sb-ND@C was then obtained by carbonizing polymer gel under an Ar/H_2_. Sb nanodot was uniformly dispersed in the carbon frameworks via in situ reduced, owing to uniform dispersion of Sb^3+^ confined in poly(ionic liquid) (PIL). As a contrast, the quality retention of Sb-ND@C without adding SbCl_3_ (denoted as C) while carbonized under the same condition was evaluated.

According to the Transmission electron microscope (TEM) image (Figure 1b), Sb nanodots were uniformly dispersed in the carbon frameworks, as well the average diameter was about 5 nm. Moreover, high-resolution transmission electron microscope (HRTEM) characterization (Figure 1c) was performed to observe the feature, where the interplanar spacings of 2.2 Å was assigned to the (104) plane of the Sb nanodot. The selected area electron diffraction (SAED) pattern in the inset of Figure 1c indicated the polycrystalline structure of Sb-ND@C and coincides with the typical hexagonal Sb phase. As displayed in Figure 1e, the surface of Sb-ND@C appears smooth, indicating that the antimony particles were embedded in the carbon frameworks and too small to be observed. According to the energy-dispersive X-ray spectroscopic (EDS) mapping of the SEM image, Sb elements were uniformly distributed in the carbon frameworks.

The crystal structure of Sb-ND@C was investigated by the X-ray diffraction (XRD). As shown in Figure 1d, the diffraction peaks in the XRD pattern of the Sb-ND@C material were indexed to hexagonal Sb (JCPDS No. 35-0732), without any other phases or impurities, indicating that Sb^3+^ was reduced completely to metallic Sb during the synthesis process. Based on the Scherrer Equation for the (012) peaks of Sb, the average size of Sb particles was calculated at around 3.2 nm. No diffraction peaks of carbon materials were detected, indicating the amorphous structure of the carbon frameworks. The XRD patterns and SEM images of Sb-RGO, Sb and C were shown in Appendix A.

The surface chemical composition of Sb-ND@C was characterized by X-ray photoelectron spectroscopy (XPS). In the survey XPS spectrum (Figure 2a), the corresponding characteristic peaks of each element confirmed the presence of C, P, N and Sb elements. The inset of Figure 2a displays three characteristic peaks at 395.6, 398.57 and 401.27 eV corresponding to pyridine nitrogen, pyrrole nitrogen and graphite nitrogen, respectively [22,23,24]. As shown in Figure 2b, three characteristic peaks can be observed at 285.59, 284.78 and 289.10 eV indicating C=N, C−C and C=O [22,25,26,27]. Figure 2c can be deconvoluted into two peaks, P 2p_3/2_ (133.85 eV) and P 2p_1/2_ (134.62 eV) [28]. Studies confirm that nitrogen doping can generate more chemically active sites and more electrons with the phosphorus-conjugated carbon frameworks to improve the electronic conductivity of the electrode. Figure 2d shows the high-resolution spectra of antimony. The characteristic peaks at 531.4 and 541.09 eV correspond to Sb 3d_5/2_ and Sb 3d_3/2_ [22], respectively. The characteristic peak appearing at 532.14 eV corresponds to the characteristic peak of oxygen, which may be caused by the adsorption of water vapor from air by the material during the characterization process [29].

As illustrated in Figure 3a, Fourier transform infrared (FT-IR) spectrum of Sb-ND@C confirmed the presence of N−H (3446 cm^−1^), C=N (1561 cm^−1^), and C−N (1094 cm^−1^) bonds in the carbon frameworks. The peak (2360 cm^−1^) was related to asymmetrical stretching vibration of CO_2_ in the air. Compared to C, when Sb-ND is loaded, the C−N stretching vibration band of Sb-ND@C was weakened and the corresponding absorption peak shifted to lower wave number of 1040 cm^−1^. This means that there is a possible formation of Sb−N−C bonds, which weakens the C−N bond and may improve the stability of Sb-ND [30]. The Raman spectra of Sb-ND@C and C were shown in Figure 3b. The intensity ratio of the D band and G band (I_D_/I_G_) was used to evaluate the degree of carbon defects. Compared to C, the I_D_/I_G_ of Sb-ND@C increased from 1.94 to 2.17 obviously, indicating the increase of defects between the interfaces of Sb and carbon supports after Sb-ND was doped. Then another discovery was that the D band of Sb-ND@C was shifted from 1386 to 1378 cm^−1^ compared with C, which also implied an interaction between Sb and C atoms. The loading of metal sites on the carbon defects could be considered as the probable active sites during the electrochemical reaction [31].

The thermogravimetric (TG) analysis was conducted in the air to evaluate the mass ratio of Sb in Sb-ND@C. As shown in Figure 3c, the TG curve of Sb-ND@C was horizontal gradually after 800 °C, which meant that C had been completely converted into CO_2_ and Sb had been oxidized into Sb_2_O_4_ [32]. Based on the total weight loss of 25.4%, it can be calculated that the mass ratio of Sb in Sb-ND@C was 58.7%. The nitrogen adsorption/desorption isotherms indicated that BET surface area of the Sb-ND@C sample was 38.7 m^2^ g^−1^ and the mean pore diameter was about 6.4 nm which is basically consistent with TEM and XRD (Figure 3d).

The electrochemical performances of Sb-ND@C and reference samples as the KIBs anode were investigated in a voltage window of 0.01–2.0 V (vs. K^+^/K) in Figure 4. CV of the Sb-ND@C anode at a scan rate of 0.05 mV s^−1^ for the first five cycles are shown in Figure 4a. The first cathodic scan is dissimilar to the following scans, implying activation process and solid electrolyte interface (SEI) film formation between electrode and electrolyte during the first cycle. Specifically, the cathodic peaks at 0.5 and 1.0 V correspond to electrochemical reaction of Sb to form amorphous K_x_Sb and then cubic K_3_Sb alloy phase. While for the anodic sweep, a broad oxidation peak from 0.1 V to 0.25 V may be ascribed to depotassication reaction of the K_x_Sb alloy [13,33,34]. Moreover, lower cathodic peak is detected at 1.04 V in the first potential sweep process, corresponding to the formation of SEI film [17]. In addition, the overlapping CV curves from the second cycle indicating that the Sb-ND@C electrode possesses highly reversible electrochemical performance.

Figure 4b displays the 1st, 2nd and 100th cycles of galvanostatic charge and discharge profiles for the Sb-ND@C electrode at 100 mA g^−1^. The Sb-ND@C delivers an initial discharge/charge capacity of 869.2/591.1 mA h g^−1^ with 68.0% Coulombic efficiency. The loss of capacity should be attributed to the formation of SEI film on the electrode/electrolyte interphase, which is in conformity with the CV profiles. In addition, the galvanostatic charge and discharge profiles for the 2nd and 100th cycles tend to overlap, demonstrating a satisfactory reaction reversibility.

The rate performance of the Sb-ND@C electrode was further evaluated at various current densities from 0.05 to 5 A g^−1^, as shown in Figure 4c. As can be observed, the Sb-ND@C electrode delivered specific capacities of 641.0, 610.1, 580.2, 541.0, 492.3, 413.1 and 320.1 mA h g^−1^ at 0.05, 0.1, 0.2, 0.5, 1, 2, and 5 A g^−1^, respectively. Additionally, when the current density return to 0.05 A g^−1^, the capacity can recover to 630 mA h g^−1^. The excellent rate capability of the Sb-ND@C electrode may be associated with the homogeneous distribution of Sb-Nanodot encapsulated in the Sb-ND@C electrode and the good electrical conductivity of the amorphous carbonaceous layers.

The comparison of the cycling performance of the Sb-ND@C, Sb/rGO, Sb and C electrodes at 500 mA g^−1^ was shown in Figure 4d. Impressively, the Sb-ND@C delivers a capacity as high as 486.7 mA h g^−1^ even after 1000 cycles, achieving 89.8% capacity retention. For bare Sb electrode, its capacity decayed rapidly after the initial several cycles, and only retained 24.3 mA h g^−1^ after 100 cycles, which could be related to the structure destruction of the bare material under high current density. Furthermore, the cycling performance of carbon frameworks (without the addition of SbCl_3_) exhibits a reversible capacity of 110 mA h g^−1^ in the first 100 cycles, which is attributed to the low potassium storage properties of carbon materials. Besides, as a comparison, the Sb/rGO electrode is synthesized with the same Sb contents and annealing temperature. The cycling performance of the Sb/rGO electrode exhibits 260 mA h g^−1^ after 45 cycles and 31.1 mA h g^−1^ after 500 cycles which is relatively lower and decreases rapidly than Sb-ND@C. As shown in Appendix A, we also provided the galvanostatic discharge–charge profiles for the 1st, 2nd and 100th cycles of Sb/rGO, Sb and C electrodes. The cycling performances of the Sb-ND@C, Sb/rGO, Sb and C electrodes agree with the Figure 4d. Moreover, cycling performance of our Sb-ND@C electrode was compared with the state-of-the-art ones previously reported as shown in Appendix A suggesting the superior K^+^-ion storage capability of the Sb-ND@C composite. Owing to the advantages of the in situ confined polymerization strategy, the contents and distribution of Sb in Sb-ND@C were nearly identical which improves the utilization rate of active substances and mitigates volume change.

The outstanding charge-discharge characteristics of Sb-ND@C can be attributed to the perfect electrochemical reaction pathways as shown in Figure 5. Firstly, the introduction of N and P atoms further generated more electrochemical reactive sites and promotes electron transfer during the electrochemical reaction. Secondly, Sb-nanodot robustly attached on carbon frameworks by in situ confined polymerization of ionic liquids, which enhanced intimate contact and provide continuous electron transport pathways. Thirdly, uniformly dispersed and nano sized Sb particles in carbon frameworks further shorten the ionic diffusion path, resulting in accelerating K^+^ diffusion in Sb. Additionally, the unique porous architecture accommodates the strain induced by the volume variation during discharge/charge procedure and favor electrolyte penetration which facilitating fast K^+^ migration between electrode and electrolyte. All of these impacts are beneficial for effectively accelerating electron transfer and potassium-ion diffusion, enabling outstanding high-rate performance and excellent cycling stability.

## 4. Conclusions

In summary, we have prepared uniformly dispersed Sb-Nanodot in carbon frameworks via in situ confined polymerization of ionic liquids. The synthesized Sb-ND@C composite as an anode for KIBs exhibits a high reversible capacity of 591.1 mA h g^−1^ at 100 mA g^−1^, a superior rate performance of 320.1 mA h g^−1^ at 5 A g^−1^, and improved cycling stability of 486 mA h g^−1^ after 1000 cycles, achieving 89.8% capacity retention. Outstanding electrochemical properties of Sb-ND@C are attributed to the homogenous distribution of Sb-Nanodot in Sb-ND@C by in-situ polymerization strategy, which not only buffers the volume change but also avoids the agglomeration of antimony in the process of charge and discharge. This provides a promising approach for the designing and fabricating of uniformly dispersed metallic nanodot with outstanding performance for electrode material.

## Data Availability

The data presented in this study are available on request from the corresponding author.

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
