# Peer review of "Uniformly Dispersed Sb-Nanodot Constructed by In Situ Confined Polymerization of Ionic Liquids for High-Performance Potassium-Ion Batteries"

_molecules, 2023, doi:10.3390/molecules28135212_

Round 1

Reviewer 1 Report

This paper describes the preparation of a Sb-nanodot in carbon framework, its characterization and electrochemical performance in K half-cell.

This paper could be of interest but needs some revision before being considered for publication:

It is necessary to indicate the electrode loading of the investigated material. It is almost impossible to appreciate its performance without this data.

The experimental part, electrochemical performance, please replace lithium foil by potassium.

Figure 1. b is difficult to read. The Sb size distribution should be put in a separate window.

Figure 3.d the pore size distribution looks very broad and not really consistent with the claims in the text.

There is no mention of the mechanism of potassium insertion/extraction from Sb. It could be necessary to explain why in the CV the peaks around 1V vs K/K+ does not appear reversible.

Please insert voltage vs K/K+ in figure 4.b

As mentioned previously it is pointless to compare materials performance if their respective loading is not mentioned. Please add this information to the manuscript.

English is fine but few typos are detected such as:

annealing instead of anealing in scheme 1

potassiumation/depotassiumation in figure 5.  potassiation/depotassiation is used in the caption of the same figure. 

Reviewer 2 Report

The manuscript titled “Uniformly Dispersed Sb-Nanodot Constructed by In Situ Confined Polymerization of Ionic Liquids for High-Performance Potassium-Ion Batteries” describes the preparation of Sn nanoparticles in ionic liquid based carbon materials. The manuscript can be accepted after addressing the following comments.

Scheme-1, the comparison with the literature is confusing. It is not a correct method to categories in a single step.

It is not clear, addition of carbon black in the paste. The content of the binder is 80% which is quite high.

Figure 1a and scheme 1. It is the duplication of the schematic presentation.  

It is not clear, the authors are interested to develop anode or cathode. The cell voltage from the Figure 4 is very low.

It is really difficult to understand the formation of KxSb alloy. There is no spectroscopic evidence. It is also possible that intercalation of K+ into the carbon material/conducting carbon black. The contribution from the carbon material/conducting carbon black needs to be calculate.

Check for the typos.    

NA

Reviewer 3 Report

This work is interesting and the results are well organized with the relevant discussion, but here listed a few comments for improving the manuscript & better reach.

  1. Figure 2, Figure 3 & Figure 4 images quality should be improved.

2. In Figure.1 Authors provided qualitative characterizations for all the test samples but suggesting authors provide similar comparative data for all the test samples for Figure 4 for comparison. ( if not in the main context results and discussion provide its comparison studies in supplementary data).

3. In Figure 4, test batteries cell performance was provided but not coulombic efficiencies, suggesting adding their coulombic efficiencies and their description for further reference.

4. In Figure 4b, suggesting authors provide Galvanostatic discharge–charge profiles for the first 3 cycles of all the test samples to compare and analyze their initial cycle performances.

5. In Figure 4d Sb-ND@C sample cell shows excellent performance over other test samples but it's interesting to know the detailed explanation for their initial cycles capacity fade and stabilization. A necessary detailed explanation and mechanism should be added.

6. Suggesting authors mention their test cells mass loading to compare their cell performance.

7. As the authors mentioned their Sb-ND@C cell has excellent and stable performance, it's better to provide tabulated literature references and their mass loading and cell performance of the metal composites in a similar field.( To support significant results)

Round 2

Reviewer 1 Report

The authors addressed the different questions and remarks related to this manuscript. This paper can now be published in Molecules.

Reviewer 3 Report

The authors revised the manuscript and well responded to the comments. Updated all the changes and highlighted them in this version.